# Using Direct Solar Energy Conversion in Distillation via Evacuated Solar Tube with and without Nanomaterials

Bahaa Saleh [1,*], Fadl A. Essa [2,*], Zakaria M. Omara [2], Mohamed H. Ahmed [3], Mahmoud S. El-Sebaey [4], Mogaji Taye Stephen [5], Lingala Syam Sundar [6], Mohammed A. Qasim [7], Eskilla Venkata Ramana [8], Sengottiyan Shanmugan [9] and Ammar H. Elsheikh [10]

1    Department of Mechanical Engineering, College of Engineering, Taif University, Taif 21944, Saudi Arabia
2    Mechanical Engineering Department, Faculty of Engineering, Kafrelsheikh University, Kafrelsheikh 33516, Egypt; zm_omara@yahoo.com
3    Mechanical Engineering Department, King Abdulaziz University, Jeddah 21589, Saudi Arabia; mhaahmed1@kau.edu.sa
4    Mechanical Power Engineering Department, Faculty of Engineering, Menoufia University, Shebin El-Kom 32511, Egypt; eng_mahmoudelsebaey@yahoo.com
5    Department of Mechanical Engineering, School of Engineering and Engineering Technology, Federal University of Technology Akure, Akure 340110, Nigeria; tsmogaji@futa.edu.ng
6    Department of Mechanical Engineering, College of Engineering, Prince Mohammad bin Fahd University, Alkhobar 31952, Saudi Arabia; slingala@pmu.edu.sa
7    Nuclear Power Plants and Renewable Energy Sources Department, Ural Federal University, 620002 Yekaterinburg, Russia; s.e.scheklein@urfu.ru
8    Department of Physics, University of Aveiro, I3N-Aveiro, 3810-193 Aveiro, Portugal; venkataramanaesk@rediffmail.com
9    Research Centre for Solar Energy, Department of Engineering Physics, College of Engineering, Koneru Lakshmaiah Education Foundation, Guntur 522502, India; s.shanmugam1982@gmail.com
10   Department of Production Engineering and Mechanical Design, Tanta University, Tanta 31527, Egypt; ammar_elsheikh@f-eng.tanta.edu.eg
*    Correspondence: b.saleh@tu.edu.sa (B.S.); fadlessa@eng.kfs.edu.eg (F.A.E.)

**Abstract:** As is widely known, the issue of freshwater scarcity affects practically all people, and all are looking for innovative and workable ways to attempt to solve this issue. In this work, a novel method of desalination is proposed. The proposed system consists of a solar collector (PTSC), evacuated pipe (EP), condenser (CU), and separation unit (SU). The working principle of the system is heating the feed saline water using the PTSC and EP and controlling the water flow rate to control the output conditions of the EP. The produced vapor is therefore separated from salty water using the SU. In addition, the generated steam is condensed into the CU to produce a freshwater distillate. Consequently, the effect of solar radiation on the affecting temperatures was tested. In addition, the effect of using different water flow rates (6, 7.5, 10, 20, 40, and 60 L/h) inside the EP on the system productivity was investigated. The primary findings of this work may be highlighted in relation to the experiments conducted. At midday, when ultraviolet irradiance reached its highest, the EP's water flow entrance and outflow had the largest temperature differential. In addition, the lower the water flow rate inside the EP, the higher the water temperature, the higher the evaporation rate of the system, and the greater the freshwater productivity of the system. At 6 L/h, the water's highest temperature was 92 °C. Moreover, the best performance of the system was obtained at 7.5 L/h, where the freshwater production and average daily effectiveness of the distillate process were 44.7 L/daytime and 59.6%, respectively. As well, the productivity of EP was augmented by around 11.86% when using graphite nanoparticles. Additionally, the distilled freshwater from the system operating at the flow rate of 7.5 L/h costs 0.0085 $/L.

**Keywords:** solar collector; separation; evacuated pipe; condensation; direct immediate distillation

## 1. Introduction

Without water on the earth, there would be no life for all human beings. Moreover, clean water is an essential working fluid of many industrial and medical processes and applications. Despite the importance of clean water as previously explained, there is only 2% of the water on the earth's surface that is drinkable, or it can be called clean water [1,2]. Therefore, humanity has no choice but to address this problem by finding appropriate solutions to transform salt and brackish water into clean drinking water [3–5]. As a result, today, we have two main approaches for purifying water to obtain potable water: one is a commercial approach such as MSF, RO, and membrane distillation. The other approach is used for the personal use of small families such as solar distillation [6–8]. Although solar distillation is the simplest option for purifying saline water, it suffers from low freshwater production [9–11]. Consequently, scientists have introduced various solutions to overcome this limitation of low productivity [12]. Innovations to the layout and functioning of solar desalination processes were among the answers [13–16]. As a result, the experts converted distillers, which are within the desalination schemes, to stepped (distiller with vertical steps to increase surface area) [17–19], disc (distiller with rotating discs to increase the surface area and break surface tension) [20], vertical solar distillers [21], tube-shaped (tubular distiller to enhance evaporation and condensation) [22–24], drum (distiller with rotating cylinders inside it) [25–27], photovoltaic thermal [28,29], fins (distiller with longitudinal fins) [30,31], trays (distiller with suspended trays) [32–34], tilted [35], woven materials [36–38], zig-zag (distiller with corrugated surfaces to improve evaporation) [39–41], spheric [42,43], double-stage [44–46], multi-stage [47], inverted absorber [48], convex [49,50], and pyramid [51–53]. Furthermore, various changes in the operating conditions such as using condensers [54–56], nanofluids [56–58], nanocomposites [59–63], thermal exchangers [64], drifting aluminum sheets [65], desiccant [66], solar pools [2], cooling cover [16], volcanic stones [67], woven ropes [41], rotational compartments [68–70], nano-coated surfaces [71,72], phase-change materials (PCM) [73–75], magnets (distiller with magnets to improve vaporization) [31,76], sun ray collectors [77], solar trackers [78], multi-stage basins [79], mirrors (distiller with reflectors to focus solar rays on basin water) [80], and recycling vaporization latent heat [81] were proposed as significant modifications. All of these amendments were proposed to improve the thermal and economic performances of the solar still as a member of the solar desalination systems. On the other hand, there were the multi-stage flashing, multi-effect boiling, and reverse osmosis methods which produce high amounts of freshwater distillate, but they consume high power.

A tool used to focus the sun's radiation into a path of action is the parabolic trough solar collector (PTSC). As a result, the feedwater for the solar distillers was warmed using it. For instance, Aboelmaaref et al. [82] surveyed the many uses of PTSC and dish concentrators in desalination processes driven by renewable power. An osmotic purification device integrated with PTSC and a wind turbine was proposed by Makkeh et al. [83]. Thermal energy was produced by PTSC and then used in the Rankine cycle to provide electricity. The reverse osmosis device was powered by the produced electricity. This configuration resulted in a 23% reduction in the price of the acquired freshwater. Additionally, this technology reduced $CO_2$ emissions by 52,164 tons annually. In order to provide drinkable freshwater, Mosleh et al. [84] incorporated a heat pipe and a dual-glass evacuating tube collector. They achieved an efficacy of 65.2% and a yield of 0.933 kg/(m$^2$·h). A multi-stage flashing, a form of sun-purification processes driven by PTSCs, had its life-cycle cost examined by Ziyaei et al. [85]. They came to the conclusion that the systems employing the sun's radiation and natural gas in Hurghada, Egypt, where 88% of the needed power was delivered by the sunlight, experienced the least cycle costs. The least-benefited cities from sunlight were Manzanillo and Los Angeles, with 59% and 57%, correspondingly. The PTSC was utilized by Narayanan and Vijay [86] to lower the pH of saltwater. According to the operating temperature of the moving salty water, they reported a pH of 7.301–7.5878.

Regarding the above literature, a novel method of desalination is proposed. The proposed system consists of a parabolic trough solar collector (PTSC), evacuated pipe (EP),

condensation unit (CU), and separation unit (SU). We used the commercial EP which is cheap and available in the market. The working principle of the system is heating the feed saline water using the PTSC and EP and controlling the water flow rate to control the output conditions of EP. The produced vapor is then separated from saltwater using the SU. In addition, the generated steam is condensed into the CU to produce a freshwater distillate. Consequently, the effect of solar radiation on the affecting temperatures was tested. In addition, the effect of using different water flow rates (6, 7.5, 10, 20, 40 and 60 L/h) inside the EP on the system productivity was investigated. In addition, the effect of using graphite nanoparticles mixed with the feed water of the EP is investigated. The nanoparticles would improve the heat transfer characteristics of nanofluids which enhances the performance of the desalination system.

## 2. Methodology

### 2.1. Fabrication of Test-Rig

#### 2.1.1. Setup Components

The test rig (shown in Figure 1) has four main parts: feed water tank (FWT), parabolic trough solar collector (PTSC), separation unit (SU), and condensation unit (CU). The FWT was of circular shape with 50 cm diameter and 50 cm length. Additionally, the PTSC is a parabolic with the specifics shown in Figure 2. Table 1 contains the specifics of the PTSC structure's specifications. We used the same dimensions obtained from Ref. [84] for reliable results and comparison. As seen in Figure 3, the PTSC's structure was constructed from wood to save costs, mass, and complexity. A 201-corrosion resistance steel was used to create the reflecting body (0.4 mm sheet). By use of a metallic support, the PTSC was fastened. It can rotate the entire PTSC, which allows it to follow the sun.

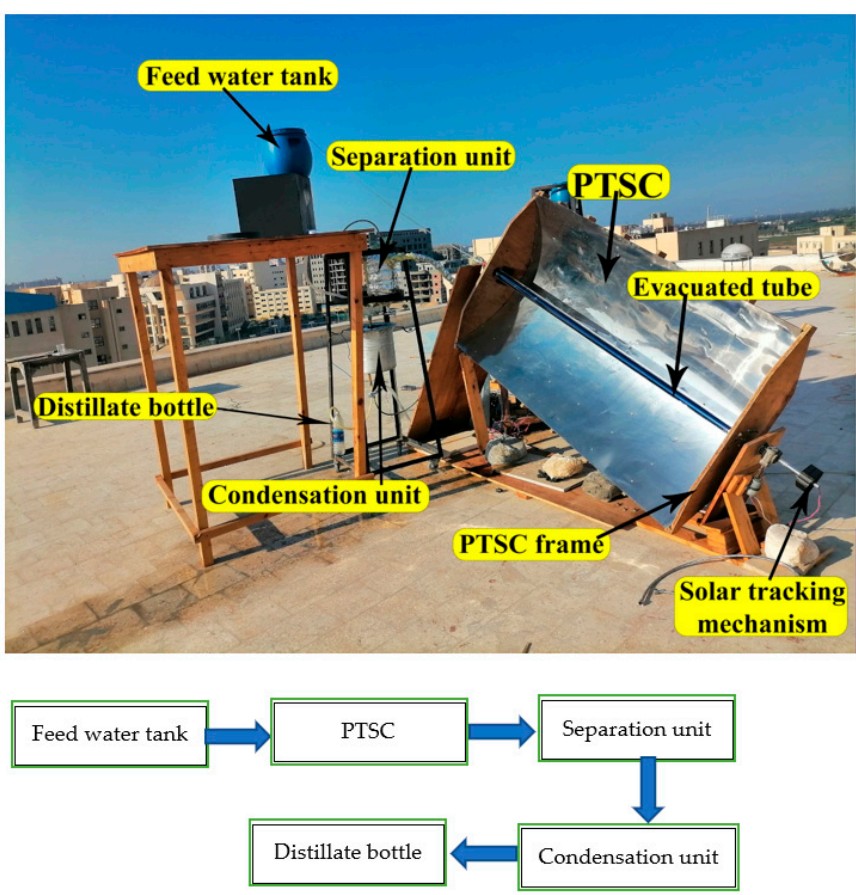

**Figure 1.** Photograph of the test-rig.

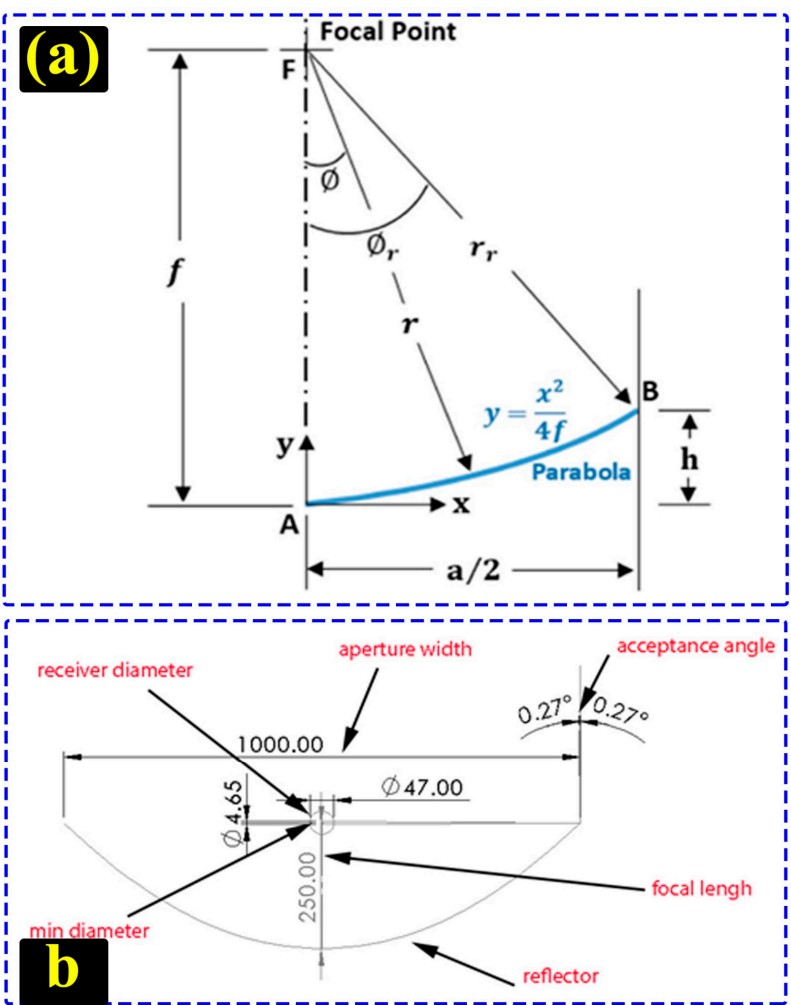

**Figure 2.** Design of PTSC: (**a**) for the focal length and (**b**) for the reflector.

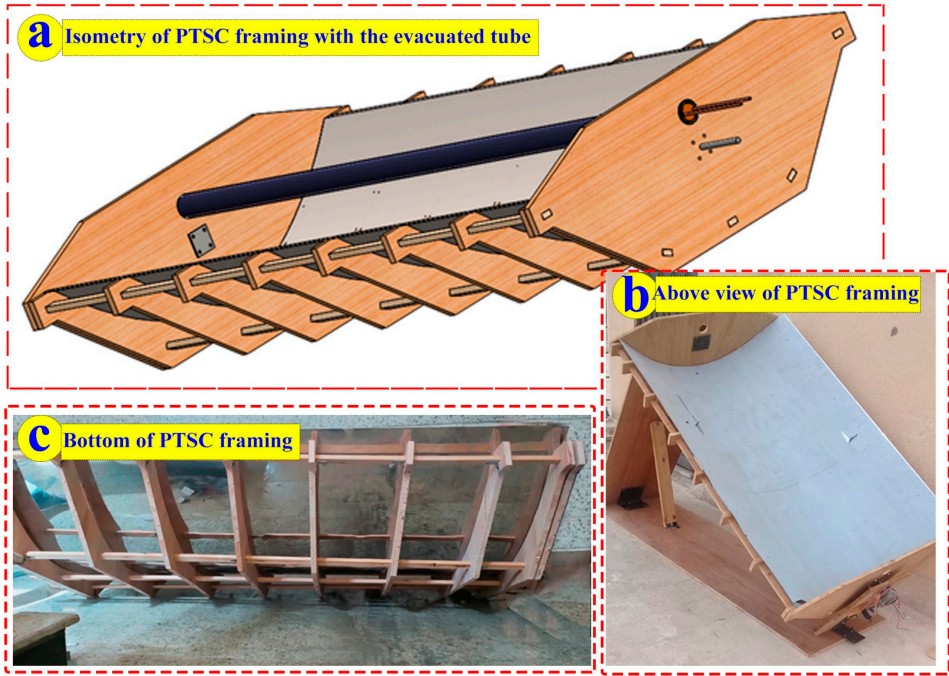

**Figure 3.** PTSC framing and EP.

**Table 1.** The specific PTSC specifications [84].

| Parameter | Description | Value |
|---|---|---|
| $\phi_r$ | Rim angle | 90° |
| $w$ | Aperture of parabola (width) | 1 m |
| $f$ | Focal height | 250 mm |
| $r_r$ | Rim radius | 500 mm |
| $D_{min}$ | Minimum diameter of receiver | 4.65 mm |
| $C$ | Concentration ratio | 6.77 |

2.1.2. The Absorber Pipe (Evacuated Pipe) of PTSC

Two concentric cylindrical tubes (metallic internally and glass externally) were used to create the evacuation pipe (EP), as illustrated in Figures 3 and 4. In between the two tubes, there existed a suction film (outer and inner tubes). Suction has the ability to reduce heat losses and boost performance. The evacuated absorbing tube had exterior and interior diameters of 58 mm and 47 mm, respectively. It was 1.8 m long. A copper U-pipe was utilized within the tube to circulate the liquid. Since the inner surface of the EP is extremely hot and the metal tubing has a cold surface, a direct connectedness between both interfaces would indeed damage the glass tubing, erasing the EP's suction property. Consequently, an insulation for the copper tube is made using two different materials (aluminum foil and transformer cooling oil). The evacuated pipe is boosted with a transformer oil mixed with graphite nanomaterials. The function of nanomaterial is to enhance its heat transfer characteristics. Additionally, the properties of nanomaterials are tabulated in Table 2.

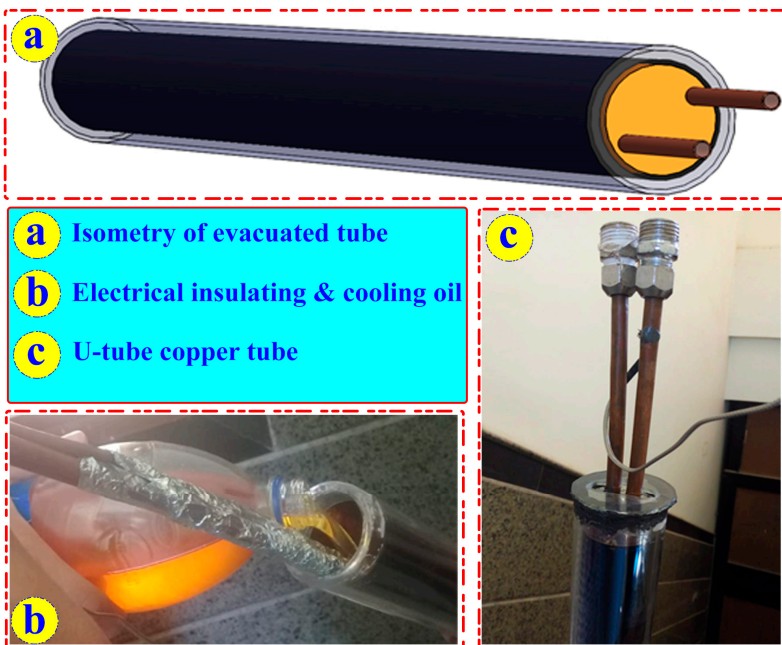

**Figure 4.** Evacuated pipe of PTSC.

**Table 2.** Properties of graphite nanomaterials.

| Property | Value |
|---|---|
| Density | 1.8 g/cm$^3$ |
| Thermal conductivity | 195 W/m °C |
| Thermal expansion coefficient | 3.2 μm/mK |
| Porosity | 13% |

### 2.1.3. Sun Tractor System

A vast, flexible network of particular axis-monitoring solar collectors makes up the solar beam of a concentrated solar power station. Such collectors are arranged in several parallel lines over the sun field, often oriented on a horizontal axis running from north to south. As shown in Figure 1, a mono-axis tracking device was utilized to point solar collectors and thermal recipients toward the sunshine. To optimize the gathering of radiation, the PTSC was often orientated North–South and tracked the sunlight as it moved from East to West. A motor (DC), speed restriction type, optical sensitivity spectroscopic techniques, and microcontroller made up the tracker control scheme. Different optical resistances were mounted on each side of the PTSC mirror to make up the photo resistance detector module. This optical impedance sensor module is in charge of generating a voltage proportionate to variations in the incidence angle of the sunrays on the photographs; the microcontroller then translates this indication into rotary motion until the potential drops to zero once more to halt the rotation.

### 2.1.4. Condensation Unit

The temperature difference between evaporating and condensing areas in the desalination system affects its productivity. Studies stated that raising the evaporator–condenser temperature difference enhances the desalination system productivity. Then, the outside condensing component was included into the arrangement (CU). It was made up of two sealed, circular metal containers that store coolant used to condensate steam within a copper helix. The CU had 4 openings: 2 for the cold fluid, one for the distillate liquid, and one for the mist. The specifics of CU are depicted in Figure 5.

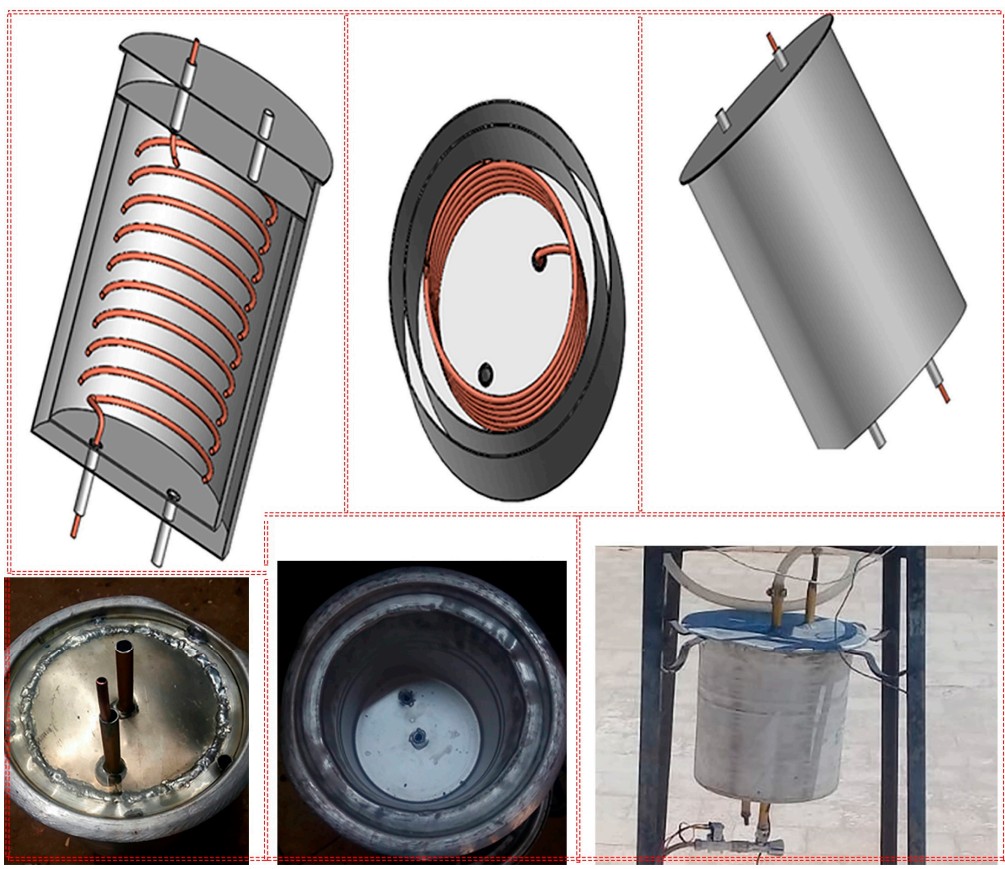

**Figure 5.** Different details of the condenser investigated.

### 2.1.5. Separation Unit

As shown in Figure 6, a separating component (SU) is a mechanism used to extract hot water from steam that is produced. High-temperature water comes in the SU from the PTSC. To the separator from the point of entrance, the temperature might rise to 99 °C. Below is an explanation of the system's operation working. In vapor–liquid separators, density is the key to working mechanism. It is used to move the less dense fluid (mist) to the top of the container and the denser fluid (water) to drop to the bottom, in which it is extracted. The tubes that draw the steam were insulated with fiberglass to avoid condensing on them and to maintain a consistent temperature for the water and steam within the separator (SU).

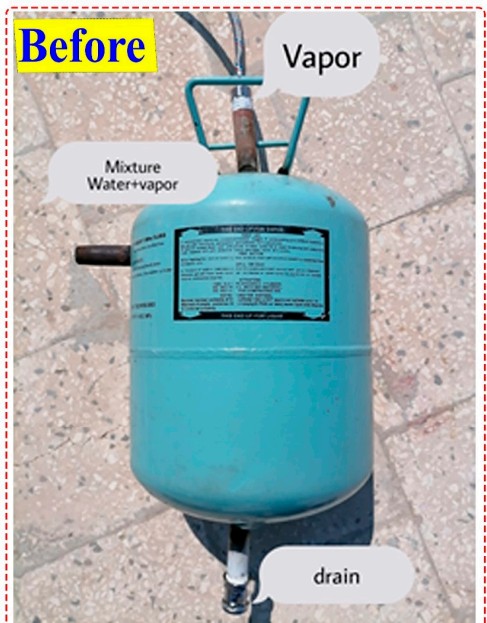
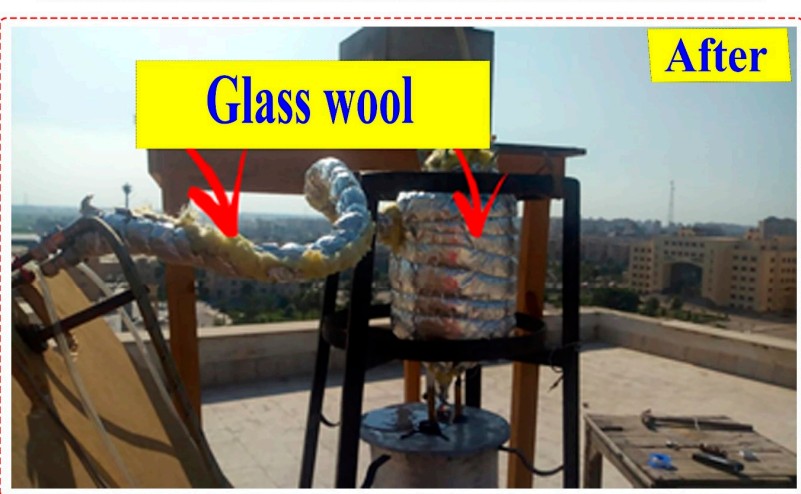

**Figure 6.** Separator (SU) installed in the proposed system both before and after installation.

### 2.2. Used Measurable Instruments

The specifications of used measurable tools are summarized in Table 3. The temperature readings at the water, surrounding air, vapor, and condensing points are reported using temperature sensors and thermocouples. The aforementioned detectors are linked to an Arduino device to translate the analog signals to digital values. The solar power meter also recorded the luminosity. Additionally, the air velocity was set using a GM8908 wind velocity measurement instrument, and a sensitivity scale (flasks) was used to indicate the distillation level.

**Table 3.** Specifics of used measurable tools.

| Instrument | Dimension | Unit | Resolution | Accuracy | Range | Error |
|---|---|---|---|---|---|---|
| Solarimeter | Solar intensity | W/m$^2$ | 0.1 | ±1 | 0–5000 | 1.5% |
| Waterproof temperature sensors of ds18b20 | Temperature | °C | 0.1 | ±0.5 | 55–100 | 1% |
| DHT11 temperature sensor | Temperature | °C | 0.1 | ±1 | 0–50 | 1% |
| K-type thermocouple | Temperature | °C | 0.1 | ±0.5 | 0–100 | 1.2% |
| Wind speed measuring device GM8908 | Wind velocity | m/s | 0.01 | ±0.1 | 0.4–30 | 1.2% |
| Calibrated flasks | Distillate | L | 0.005 | ±0.2 | 0–2.5 | 1.1% |

## 3. Results and Discussions

### 3.1. Effect of Solar Radiation

To be able to evaluate the performance of the system, the solar radiation, air temperature, inlet water temperature, outlet water temperature, and distillate quantity were measured every hour from sunrise to sunset. To ensure the reliability of the results, each experiment was repeated twice, and the average was taken between them. We preferred to display the results of these parameters at the flow rate of 7.5 L/h because this flow rate provided the best performance of the system as will be explained in the next section. Moreover, the figures were set at one flow rate only to prevent repetition in the results and unjustified exaggeration, because the results are almost similar, but with a difference in values only. So, Figure 7 illustrates the luminosity, air temperature, inlet and outlet water temperature at 7.5 L/h (testing day: 7 November 2021). As expected, the solar radiation begins from zero (at the second exactly before the sunrise) and increases gradually to peak at 11:00–13:00, as shown in Figure 7. After that, it begins to be decreased in the afternoons to reach its lowest at sunset. Figure 7 obtains that the highest value of solar radiation was 1150 W/m$^2$ at 12:00. As a result, the solar radiation is related to the time of the day as shown in Figure 7.

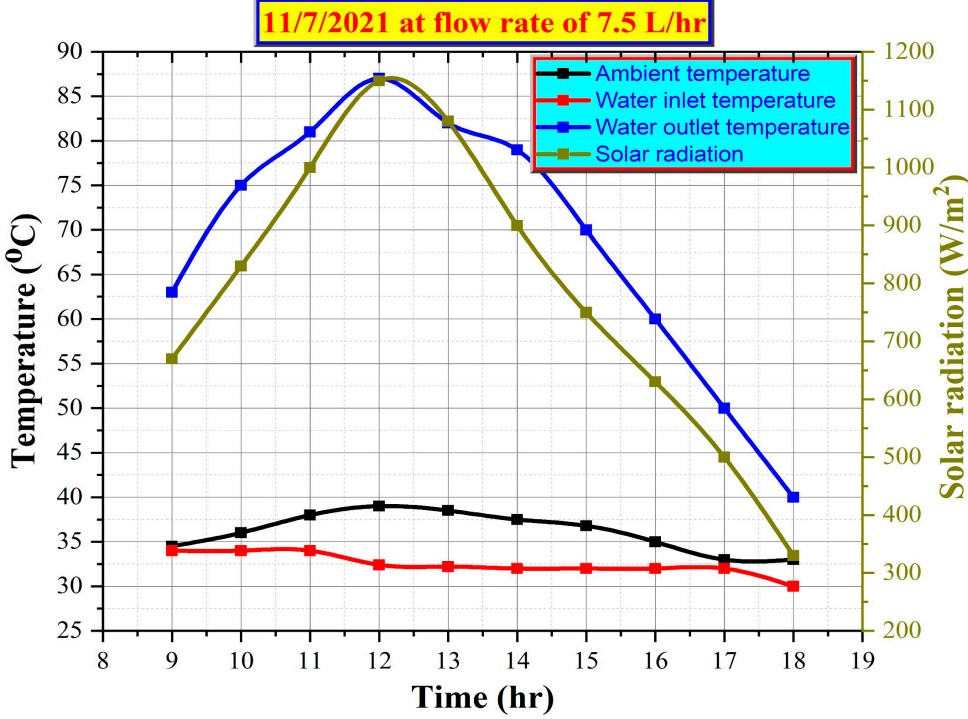

**Figure 7.** Solar radiation and air and water temperatures of the system at 7.5 L/h.

In addition, the temperatures of air and water are strongly dependent on the solar radiation as observed in Figure 7. First, the air temperature is marginally affected by the solar radiation as shown in Figure 7, where the air temperature was varied between 33 °C and 39 °C through the daytime versus the big difference in solar radiation with progressing the time. Moreover, the water temperature was strongly affected by the solar radiation as obtained from Figure 7. The inlet water temperature was almost constant through the testing day. Then, the water is passed through the evacuated pipe which lies in the focal center of the PTSC. So, the evacuated pipe is considered as the line in which the solar rays are concentrated and reflected on. As a result, during early morning, when solar irradiance was at its lowest, the water's warmth was mild. It was then steadily raised from the input level as the sun irradiation rose. For instance, the water level rose from 63 °C to 81 °C with a rise in solar energy from 670 to 1000 W/m$^2$, correspondingly, during the hours of

9:00 and 11:00. Additionally, Figure 7 shows that the greatest seawater temperature was recorded between 11:00 to 13:00 (p.m.) when sun energy was also at its highest. As a result, at 12:00, the water's highest temperature was 87 °C at 7.5 L/h, with 1150 W/m$^2$ of sun irradiance. Following that, when the solar irradiance declined throughout the afternoons, the water temperature declined, as seen in Figure 7. Therefore, it could be debunked that solar energy has a significant impact on the temperature of moving water. The effectiveness of the system is confirmed by Figure 7 even though the wind speed ranged from 0.3 to 1.5 m/s.

Additionally, Figure 8 depicts the system production and distillation operational efficiencies at 7.5 L/h. At first glance, the productivity curves had the same behavior as the curves of water temperature and solar radiation (see Figure 7). As is well known, the higher the evaporation rate of the system, the greater the freshwater productivity of the same desalination system. Raising the water temperature is important for increasing the evaporation rate, and thus the freshwater production. The curves in Figure 8 confirm that the productivity of the system increased with increasing the water temperature, where it was observed that the freshwater production was maximum at the maximum water temperature. For example, the water productivity was around 7 L/h at 12:00, where temperature of water is 87 °C. Likewise, Figure 8 depicts that the distillation processing followed the same pattern as freshwater production. The maximum instantaneous distillate process effectiveness was obtained at the highest productivity, where it was around 93.33% at 12:00. So, the distillate process effectiveness is a function of the freshwater productivity as shown in Figure 8.

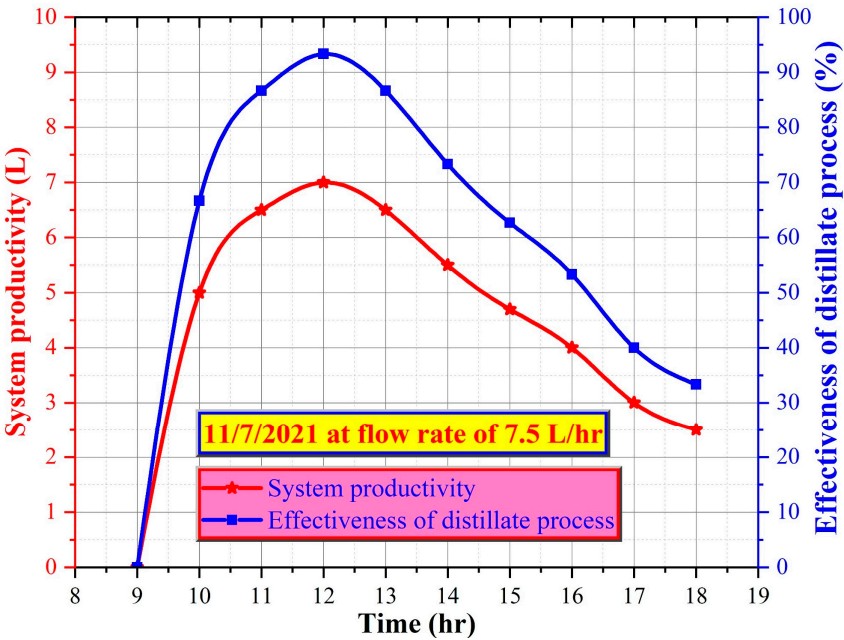

**Figure 8.** The system production and distillation operational efficiencies at 7.5 L/h.

### 3.2. Effect of Water Flow Rate

In this section, the influence of using different water flow rates (6, 7.5, 10, 20, 40, and 60 L/h) on the system performance was investigated. The distillate productivity of the system and maximum outlet water temperature under various water flow rates is presented in Table 4. In addition, Figure 9 shows the quantity of distillate and distillate process effectiveness as functions of water flow rate. In order to ensure that the comparison is impartial, and the resulting values are guaranteed and correct, the days in which the average daily solar radiation is almost equal were chosen for investigating the effect of different water flow rates on the system performance. So, the average daily solar radiation for all testing days was around ~700 W/m$^2$ as depicted in Table 4.

**Table 4.** The distillate productivity of the system and maximum outlet water temperature under various water flow rates.

| Module | Average Daily Solar Radiation | Flow Rate of Water | Maximum Outlet Water Temperature | Total Quantity of Saline Water | Distillate Productivity |
|--------|-------------------------------|--------------------|----------------------------------|--------------------------------|-------------------------|
| 1 | 720 W/m² | 6 L/h | 92 °C | 60 L/daytime | 33 L/daytime |
| 2 | 700 W/m² | 7.5 L/h | 87 °C | 75 L/daytime | 44.7 L/daytime |
| 3 | 690 W/m² | 10 L/h | 85 °C | 100 L/daytime | 58 L/daytime |
| 4 | 700 W/m² | 20 L/h | 55 °C | 200 L/daytime | 83 L/daytime |
| 5 | 710 W/m² | 40 L/h | 50.5 °C | 400 L/daytime | 60 L/daytime |
| 6 | 695 W/m² | 60 L/h | 45 °C | 600 L/daytime | 44.5 L/daytime |

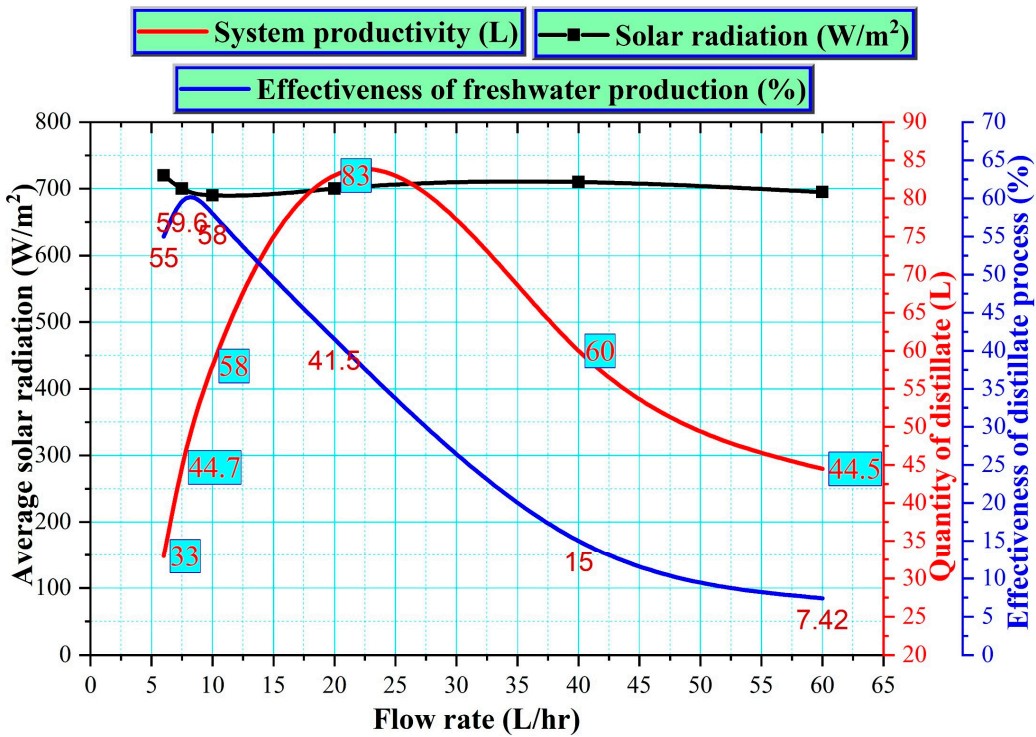

**Figure 9.** Effect of different flow rates of water on the system performance.

As is well known, the temperature of water at EP's out is raised due to lowering the flow of water within it. This is because the water has more time in the EP, which means that the water stays longer under the focused rays than the PTSC. Therefore, the maximum water temperature at the outlet of the EP pipe is greater at the low flow rates than at the large flow rates, and vice versa as obtained from Table 4. For example, the maximum water temperature was 92 °C, 87 °C, and 85 °C at the water flow rates of 6, 7.5, and 10 L/h, respectively. Then, the maximum water temperature is decreased more and more at high flow rates. for instance, the maximal water temperature is 55 °C, 50.5 °C, and 45 °C at the water flow rates of 20, 40, and 60 L/h, respectively. As well explained above, the higher the water temperature, the higher the evaporation rate, and the greater the distillate productivity. As a consequence, at a flow rate of 6 L/h, a total distillate of 33 L/daytime was purified from the total quantity of saline feed water of 60 L/daytime. In addition, 44.7 L/daytime was obtained from 75 L/daytime at 7.5 L/h. Moreover, the distillates of 58 and 83 L/daytime were obtained from 100 and 200 L/daytime at flow rates of 10 and 20 L/h, respectively. Furthermore, the quantity of productivity was reduced with increasing the flow rate due to increasing the velocity of the water inside the EP pipe for the high flow rates of 40 and 60 L/h as shown in Table 4.

### 3.3. Effectiveness of the Distillate Process and Thermal Efficiency of the System

Moreover, the effectiveness of the distillate process is illustrated in Figure 9. This parameter is calculated by;

$$Effectiveness\ of\ distillate\ process = \frac{Distillate\ productivity}{Quantity\ of\ saline\ water} \times 100 \tag{1}$$

As a result, we have two forms of the effectiveness of the distillate process: instantaneous effectiveness and average daily effectiveness. This is based on the quantity of distillate productivity substituted in the above equation. If the hourly productivity is used, then we have the instantaneous effectiveness. If the daily accumulated productivity is used in the equation, then we have the average daily effectiveness. Figure 9 shows the average daily effectiveness of the distillate process in the desalination system. It can be observed from Figure 9 that the average daily effectiveness had the maximum value of 59.6% at the water flow rate of 7.5 L/h, where there was a balance between the time the water spent exposed to the focused solar radiation on the EP pipe and the water flow rate in the pipe because with the increase in the flow rate, the productivity increases, but the effectiveness of the extracted produced water decreases because, on the other hand, we use larger quantities of water and therefore the costs are higher. This means that the best flow rate from which the largest amount of freshwater is extracted was 7.5 L per hour as illustrated in Figure 9. Other values of distillate process effectiveness are found as 55% and 58% when using the water flow rates of 6 and 10 L/h, respectively. In addition, very low effectiveness was found at the high-water flow rates. For example, the productivity process effectiveness of 15% and 7.42% was obtained when using the flow rates of 40 and 60 L/h, respectively.

On the other hand, the thermal efficiency of the system can be evaluated as following.

$$System\ thermal\ efficiency = \frac{useful\ energy\ (distillate \times latent\ heat)}{paid\ energy\ (irradiance \times area)} \times 100 \tag{2}$$

Based on the above formula, the values of thermal efficiency of the system are presented in Table 5. For instance, the thermal efficiency was higher for the lower flow rates, where it was 70% and 77% for the flow rates of 9 and 10 L/h, respectively. In addition, it was maximum at 7.5 L/h, where it was 81%. Then, the efficiency was lowered with increasing the flow rate. So, it was 55%, 24%, and 15% at the flow rates of 20, 40, and 60 L/h, respectively.

**Table 5.** The thermal efficiency of the system under various water flow rates.

| Module | Flow Rate of Water, L/h | Thermal Efficiency, % |
| --- | --- | --- |
| 1 | 6 | 70 |
| 2 | 7.5 | 81 |
| 3 | 10 | 77 |
| 4 | 20 | 55 |
| 5 | 40 | 24 |
| 6 | 60 | 15 |

### 3.4. Effect of Using Graphite Nanoparticles

The feedwater for the evacuated pipe was mixed with 2.5% graphite nanoparticles, and the mixture was utilized to feed the evacuated pipe. The main function of using nanoparticles was to enhance the heat transfer characteristics of the flowing water, and hence the vaporization rate was to be enhanced. The specific heat capacity of the graphite nanoparticles equals 706.9 J K$^{-1}$ kg$^{-1}$. Additionally, the thermal conductivity of the graphite nanoparticles is 400 W/m·K. With utilizing the nanomaterials, the maximum outlet water temperature from the EP at 7.5 L/h was 88.5 °C. As well, the distillate productivity from the system was obtained as 50 L/daytime. Therefore, the productivity of EP was augmented

by around 11.86%. In addition, the average daily effectiveness reached 66.67% at the water flow rate of 7.5 L/h when using graphite nanoparticles compared to 59.6% at the same flow rate without nanoparticles.

### 3.5. Distilled Water Cost Analysis

To find out the extent to which this proposed device can be applied, we had to undertake an economic study for this device and calculate the cost of the freshwater produced from it. The fixed costs of the system components are tabulated in Table 6. Below are the equations used to calculate the freshwater cost [84]. Some parameters such as the system lifespan and interest rate are approximated as $n = 5$ years and $i = 15\%$, respectively. Then, the capital recovery factor is calculated by:

$$CRF = \frac{i\,(1+i)^n}{(1+i)^n - 1} \tag{3}$$

also, the fixed annual cost (FAC) is

$$FAC = F\,(CRF) \tag{4}$$

additionally, the sinking fund factor (SFF) is

$$SFF = \frac{i}{(1+i)^n - 1} \tag{5}$$

in addition, the salvage value (S) is

$$S = 0.2\,F \tag{6}$$

moreover, the annual salvage value (ASV) is

$$ASV = S\,(SFF) \tag{7}$$

also, the annual maintenance costs (AMC) is

$$AMC = 0.15\,(FAC) \tag{8}$$

the total annual cost (TAC) is

$$TAC = FAC + AMC - ASV \tag{9}$$

finally, the cost of distilled water (CPL) in \$/L is

$$CPL = TAC/M \tag{10}$$

where M is the average yearly productivity of freshwater.

By solving the above equation with the help of the data in Table 6, the distilled freshwater from the system operating at 7.5 L/h costs 0.0085 \$/L.

### 3.6. Comparison between the Findings of This Work and Other Related Works

To ensure the validity of the method, the notable findings of output and the expense of distilled output are contrasted to those of prior studies discovered in the available research. Table 7 illustrates this similarity. The efficacy of the system was compared with practically all of the most common forms. Table 7 further shows that the suggested approach provides freshwater at a lower price when contrasted with the rest of the articles.

**Table 6.** Total fixed costs of the system components.

| Unit | Cost of Component ($) |
|------|----------------------|
| Iron sheets | 55 |
| Evacuated pipe | 30 |
| Supporting legs | 30 |
| Paint | 7 |
| Insulation (Fiber glass) | 15 |
| Production | 30 |
| PVC pipes | 20 |
| Tracing mechanism | 50 |
| Pipe fitting | 30 |
| Separation unit | 25 |
| Fan | 20 |
| Tanks | 30 |
| Nanoparticles | 50 |
| Copper coil | 20 |
| Total fixed cost (F) | 412 |

**Table 7.** Comparison between the findings of this work and other related works.

| Reference | Device and Modifications | Yield Improvement (%) | Yield (L/m$^2$·day) | Cost of Freshwater ($/L) |
|-----------|--------------------------|-----------------------|---------------------|--------------------------|
| Essa et al. [17] | Stepped still with trays and nanomaterials | 55 | 6.2 | 0.015 |
| Essa et al. [24] | Tubular still with rotating drum | 175 | 6.6 | 0.024 |
| Abdullah et al. [26] | Drum still with reflectors and wick | 296 | 7.25 | 0.041 |
| Alawee et al. [51] | Pyramid still with rotating cylinders and electric heaters | 214 | 9.1 | - |
| Essa et al. [52] | Pyramid still with mirrors, cooling cycle, and wick cords | 195 | 11.5 | 0.017 |
| Essa et al. [18] | Stepped still by corrugated and curved liners, nano-PCM, wick, and vapor suctioning | 170 | 7 | 0.014 |
| Abdullah et al. [25] | Drum still with condenser | 350 | 9.22 | 0.039 |
| Essa et al. [27] | Tubular still with rotating drum, nanoparticles, parabolic solar concentrator, and PCM | 218 | 6.6 | 0.024 |
| Abdullah et al. [85] | Trays still with mirrors | 95 | 4.8 | 0.021 |
| Alawee et al. [86] | Pyramid still with wick ropes | 122 | 7.9 | - |
| Manokar et al. [87] | Pyramid still with insulation conditions and 1 cm water depth. | 19.46 | 3.72 | - |
| Present study | PTSC, EP, CU, and SU | 59.6 | 44.7 | 0.0085 |

## 4. Conclusions

In this paper, a novel method of desalination is proposed. The proposed system consists of a PTSC, EP, CU, and SU. The basin idea of the proposed system is heating the saline water inside the EP which lies in the focal center of the PTSC. Then, the SU is a mechanism used to extract hot water from the steam that is produced. In addition, the generated steam is condensed into the CU to produce a freshwater distillate. The effects of solar radiation and different water flow rates on the system productivity were

investigated. The primary findings of this study may be highlighted in relation to the experiments. The maximal temperature difference between the water inlet and outlet of EP occurred at noon time, where the sun irradiation was maximal. In addition, the lower the water flow rate inside the EP, the higher the water temperature, the higher the evaporation rate of the system, and the greater the freshwater productivity of the system. The greatest water temperature was 92 °C at 6 L/h. Moreover, the best performance of the system was obtained at 7.5 L/h, where the freshwater production and average daily effectiveness of the distillate process were 44.7 L/daytime and 59.6%, respectively. As well, the productivity of EP was augmented by around 11.86% when using graphite nanoparticles. Additionally, the distilled freshwater from the system operating at the flow rate of 7.5 L/h costs 0.0085 $/L.

**Author Contributions:** Conceptualization, F.A.E. and S.S.; methodology, F.A.E. and M.S.E.-S.; formal analysis, F.A.E. and B.S.; investigation, L.S.S., M.A.Q. and E.V.R.; resources, M.H.A.; data curation, Z.M.O.; writing—original draft preparation, F.A.E., A.H.E. and M.S.E.-S.; writing—review and editing, B.S. and M.T.S.; visualization, M.A.Q., E.V.R., M.H.A. and S.S.; supervision, F.A.E.; project administration, B.S.; funding acquisition, B.S. All authors have read and agreed to the published version of the manuscript.

**Funding:** This research was funded by the Deanship of Scientific Research, Taif University, Taif, Saudi Arabia.

**Data Availability Statement:** The data presented in this study are available on request from the corresponding authors.

**Acknowledgments:** The author B. Saleh would like to acknowledge the Deanship of Scientific Research, Taif University, Taif, Saudi Arabia for funding this work.

**Conflicts of Interest:** The authors declare no conflict of interest.

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
