# Peer review of "Using Direct Solar Energy Conversion in Distillation via Evacuated Solar Tube with and without Nanomaterials"

_processes, doi:10.3390/pr11061734_

Round 1

Reviewer 1 Report

This paper proposed a system consists of a solar collector, evacuated solar tube with and without nanomaterials, condenser, and separation unit. The saline water was heated by solar energy, and the generated steam was separated using separation unit to obtain the condensed freshwater distillate in condenser unit. The effect of solar radiation, water flow rates and graphite nanoparticles mixed with the feed water were investigated. This reviewer recommends publishing this paper with the consideration of the following in the revision.

1. Solar distillation systems do not consume other energy sources and are easy to build, which has been widely studied. Nevertheless, due to the optical and thermal losses caused by bulk heating, the photothermal conversion efficiency of solar distillation is relatively low. Therefore, the novelty of this study needs to be highlighted.

2. Some important details about the experiment, such as how the vacuum degree is determined, are not adequately explained.

3. In “3.5 Distilled water cost analysis”, it is recommended to compare the cost/advantages/ disadvantages of the proposed system for distilled freshwater in this study with the cost of other technologies.

4. There are a few spelling errors, e.g., Line 60, “the most simplest option”

 There are a few spelling errors, e.g., Line 60, “the most simplest option”, need to be carefully revised.

Author Response

General comment: This paper proposed a system consists of a solar collector, evacuated solar tube with and without nanomaterials, condenser, and separation unit. The saline water was heated by solar energy, and the generated steam was separated using separation unit to obtain the condensed freshwater distillate in condenser unit. The effect of solar radiation, water flow rates and graphite nanoparticles mixed with the feed water were investigated. This reviewer recommends publishing this paper with the consideration of the following in the revision.

Response: The authors would like to thank the Reviewer for his effort and time to make our paper in a better form. Also, we corrected the manuscript considering the Reviewer comments as following.

Comment 1: Solar distillation systems do not consume other energy sources and are easy to build, which has been widely studied. Nevertheless, due to the optical and thermal losses caused by bulk heating, the photothermal conversion efficiency of solar distillation is relatively low. Therefore, the novelty of this study needs to be highlighted.

Response: Thanks for the comment. It is highlighted in the revised version of the manuscript as requested.

Comment 2: Some important details about the experiment, such as how the vacuum degree is determined, are not adequately explained.

Response: Thanks for the comment. The authors didn’t apply any external vacuum in the system. In addition, more details about the experiment are included in the revised version of manuscript.

Comment 3: In “3.5 Distilled water cost analysis”, it is recommended to compare the cost/advantages/ disadvantages of the proposed system for distilled freshwater in this study with the cost of other technologies.

Response: Thanks for the valuable comment. It is included as requested.

Comment 4: There are a few spelling errors, e.g., Line 60, “the most simplest option” , need to be carefully revised.

Response: Thanks for the comment. It is revised and corrected.

Reviewer 2 Report

Good morning;

your article presents a new study with a low cost for the distillation of waters using solar energy. But, to make it more pleasant to read with all these results obtained, it is necessary to try to add more references in the methodological and results part (85 references in the introduction and not one in the two parts mentioned above), to review the methodological part (2.1.1 to 2.1.5) because we get lost with the figures (5 figures) and finally, I think it's better to do a comparative study with other works that have been done in the same field to show the usefulness of your study, especially in the field of green economy. Good luck

Author Response

Comment 1: Your article presents a new study with a low cost for the distillation of waters using solar energy. But, to make it more pleasant to read with all these results obtained, it is necessary to try to add more references in the methodological and results part (85 references in the introduction and not one in the two parts mentioned above), to review the methodological part (2.1.1 to 2.1.5) because we get lost with the figures (5 figures).

Response: Thanks for the valuable comment. Relevant references are used in section 3.6 to be able to see the difference and advantages of the present system compared to the other articles found in the literature.

Comment 2: I think it's better to do a comparative study with other works that have been done in the same field to show the usefulness of your study, especially in the field of green economy.

Response: Thanks for the comment. It is included as requested. Please refer to Table 5 and section 3.6.

Reviewer 3 Report

Obtaining drinking water through parabolic troughs is a topic of research in the field of solar energy and environmental engineering. There are numerous recent works on this subject. Since the 2000s, work has been carried out directed towards small productions of drinking water, with mathematical tools of the process, simulation works and with abundant experimental part.

These authors have carried out experiments on the implementation of parabolic troughs to obtain drinking water through solar distillation. The work includes the description of the systems and shows a series of results of water production. However, in this work the design is not shown, the calculations for its construction and the results shown are insufficient for a complete evaluation of the proposed system. The use of nanomaterials in improving desalination is a constantly evolving research topic. However, in this work nanoparticles are included in the title but they have not been developed in the content.

Comments:

1. The introduction is not suitable for this work, since it is very general and does not focus on the content of the article. From line 58 to 75.

2. The contribution of this work is not observed. There is no theoretical development of the process, there are no design calculations for the installation. Some data are shown that are insufficient to address the flow rates foreseen in the work and the incorporation of nanoparticles on the results. For example: the data in table 1 have not been obtained from a previous calculation whose objective is the desired flow of product water. It starts from some design data without previous considerations.

3. Nanoparticles are included in the title and are only dealt with from line 288 to 299. The information indicated in the work is insufficient.

4. It has not been possible to evaluate efficiency, performance, of these facilities, etc.

5. There is no contribution regarding similar works.

6. The cost analysis has been very similar to the indicated reference. Nanoparticles are not included in this cost evaluation.

Observations:

Different flow rates are studied, but all the results are for 7.5 L/h

In section 2.2.1 there is a reference problem, line 111

For which the wind speed has been used in this work, line 179.

In section 3.1 the authors show the results for a flow rate of 7.5 L/h, because it is where they have obtained the best results. It seems to have been a trial and error experiment, with the best result being 7.5 L/h.

The numbering of the equations is not well done. For example: equation (1) is on line 267 and on line 306.

Author Response

General comment: Obtaining drinking water through parabolic troughs is a topic of research in the field of solar energy and environmental engineering. There are numerous recent works on this subject. Since the 2000s, work has been carried out directed towards small productions of drinking water, with mathematical tools of the process, simulation works and with abundant experimental part.

These authors have carried out experiments on the implementation of parabolic troughs to obtain drinking water through solar distillation. The work includes the description of the systems and shows a series of results of water production. However, in this work the design is not shown, the calculations for its construction and the results shown are insufficient for a complete evaluation of the proposed system. The use of nanomaterials in improving desalination is a constantly evolving research topic. However, in this work nanoparticles are included in the title but they have not been developed in the content.

Response: The authors would like to thank the Reviewer for his effort and time to make our paper in a better form. Also, we corrected the manuscript considering the Reviewer comments as following. Some more requested details are included such as the nanoparticles details …etc.

Comment 1: The introduction is not suitable for this work, since it is very general and does not focus on the content of the article. From line 58 to 75.

Response: Thanks for the comment. It is modified as suggested, and more relevant articles are included.

Comment 2: The contribution of this work is not observed. There is no theoretical development of the process, there are no design calculations for the installation. Some data are shown that are insufficient to address the flow rates foreseen in the work and the incorporation of nanoparticles on the results. For example: the data in table 1 have not been obtained from a previous calculation whose objective is the desired flow of product water. It starts from some design data without previous considerations.

Response: Thanks for the comment. The design details are obtained with graphs for the parts designed and fabricated by the authors such as the PTSC, while we have mentioned the details of the parts bought from the market as they are such as the evacuated tube. In addition, the data in table 1 were taken from a previous publication for reliable results and comparison.

Comment 3: Nanoparticles are included in the title and are only dealt with from line 288 to 299. The information indicated in the work is insufficient.

Response: Thanks for the valuable comment. More details about the nanoparticles are included.

Comment 4: It has not been possible to evaluate efficiency, performance, of these facilities, etc.

Response: Thanks for the valuable comment. They are added as requested. Please refer to section 3.3.

Comment 5: There is no contribution regarding similar works.

Response: Thanks for the comment. More contributions and results are added as requested regarding the reviewer comments.

Comment 6: The cost analysis has been very similar to the indicated reference. Nanoparticles are not included in this cost evaluation.

Response: Sorry for this mistake. They are added as requested.

Comment 7: Different flow rates are studied, but all the results are for 7.5 L/h.

Response: Thanks for the valuable comment. The different flow rates were studied as clearly, but the figures were set at one flow rate only to prevent the repetition in the results and unjustified exaggeration, because the results are almost similar, but with a difference in values only. However, there are some figures and tables in which results are required at all flow rates, such as efficiency and distillate effectiveness, as in Section 3.3. Also, this information is added in section 3.1.

Comment 8: In section 2.2.1 there is a reference problem, line 111.

Response: Thanks for the comment. It is corrected.

Comment 9: For which the wind speed has been used in this work, line 179.

Response: Thanks for the valuable comment. The anemometer was used to measure the wind velocity of the ambient, but we did not present it because its impact is negligible.

Comment 10: In section 3.1 the authors show the results for a flow rate of 7.5 L/h, because it is where they have obtained the best results. It seems to have been a trial and error experiment, with the best result being 7.5 L/h.

Response: Thanks for the comment. Indeed, the effect of solar radiation was investigated under various flow rates, as mentioned in this section 3.1. It turns out that the performance is similar in terms of ups and downs, but with different values. So, the results were shown for the best productivity, which is 7.5 L/h. Also, which confirms that the study is not only a trial and error experiment, is that each experiment was repeated twice and the average was taken between them. This information is added under section 3.1.

Comment 11: The numbering of the equations is not well done. For example: equation (1) is on line 267 and on line 306.

Response: Thanks for the valuable comment. The numbering of the equations is revised and corrected.

Reviewer 4 Report

In this paper, a novel method of desalination is proposed. The system is heating the feed saline water using the PTSC and EP and controlling the water flow rate to control the output conditions of the EP. The produced vapor is therefore separated from salty water using the SU.

However, there are some questions need to be clarified.

1.      Page 3, figure 1, the photograph of the test-rig is difficult to read. Please add a figure 1 (b) to draw the process flow diagram and describe how to run the test-rig?

2.      Page 3, line 111. “Error! Reference source not found.” Please correct it.

3.      Page 3, lines 112-113. “The layout arguments were chosen with the goal of maintaining high design effectiveness.” It’s confused for the readers. What does this mean?

4.      Pages 3 and 4, lines 110-111. Figure 2. Please explain the design of PTSC in detail.

5.      Page 1. Lines 29-30. A novel method of desalination is proposed. The authors should mention “novel method” in abstract.

6.      Pages 11-13. The number of equations is repeat. Please correct it.

7.      Page 11. Lines 274-275. The average daily effectiveness had the maximum value of 59.6% at the water flow rate of 7.5 L/hr. The maximum value of 59.6% of the average daily effectiveness is corresponding to a minimum value of the average solar radiation. Does the average solar radiation effect the value of effectiveness of distillate process?

8.      Page 13. Lines 315-316. The distilled freshwater from the system operating at 7.5 L/hr costs 0.0085 $/L. This cost is much lower than RO especially for small size of desalination systems. Can authors explain why?

The author explains more clearly about above questions

Author Response

General comment: In this paper, a novel method of desalination is proposed. The system is heating the feed saline water using the PTSC and EP and controlling the water flow rate to control the output conditions of the EP. The produced vapor is therefore separated from salty water using the SU. However, there are some questions need to be clarified.

Response: The authors would like to thank the Reviewer for his effort and time to make our paper in a better form. Also, we corrected the manuscript considering the Reviewer comments as following.

Comment 1: Page 3, figure 1, the photograph of the test-rig is difficult to read. Please add a figure 1 (b) to draw the process flow diagram and describe how to run the test-rig?

Response: Thanks for the comment. A process flow diagram is added to Fig. 1 as requested.

Comment 2: Page 3, line 111. “Error! Reference source not found.” Please correct it.

Response: Thanks for the comment. It is corrected.

Comment 3: Page 3, lines 112-113. “The layout arguments were chosen with the goal of maintaining high design effectiveness.” It’s confused for the readers. What does this mean?

Response: Sorry for this confusion. This sentence is removed.

Comment 4: Pages 3 and 4, lines 110-111. Figure 2. Please explain the design of PTSC in detail.

Response: Thanks for the comment. More explanation is added as requested.

Comment 5: Page 1. Lines 29-30. A novel method of desalination is proposed. The authors should mention “novel method” in abstract.

Response: Thanks for the valuable comment. It is modified as suggested.

Comment 6: Pages 11-13. The number of equations is repeat. Please correct it.

Response: Thanks for the valuable comment. They are corrected as requested.

Comment 7: Page 11. Lines 274-275. The average daily effectiveness had the maximum value of 59.6% at the water flow rate of 7.5 L/hr. The maximum value of 59.6% of the average daily effectiveness is corresponding to a minimum value of the average solar radiation. Does the average solar radiation effect the value of effectiveness of distillate process?

Response: Thanks for the valuable comment. It is seen from section 3.3 that the daily effectiveness depends mainly on the water flow rate with the solar radiation. Also, we evaluated the thermal efficiency of the system which agrees with this conclusion.

Comment 8: Page 13. Lines 315-316. The distilled freshwater from the system operating at 7.5 L/hr costs 0.0085 $/L. This cost is much lower than RO especially for small size of desalination systems. Can authors explain why?

Response: Thanks for the valuable comment. Yes, this is because of the large amount of electrical energy used by the reverse osmosis units and the presence of pumps in these units, unlike this proposal that relies mainly on solar energy. However, one of the disadvantages of this system is the low productivity compared to reverse osmosis units.

Round 2

Reviewer 1 Report

The paper can be published.

Author Response

Thanks for your suggestion about the publication of the paper.

No other action is needed here.

Reviewer 3 Report

The authors of the paper have made the appropriate modifications to the comments. Good job.

Author Response

Thanks for your feedback. No other action is needed here.